# Arginase-1+ Exosomes from Reprogrammed Macrophages Promote Glioblastoma Progression

**DOI:** 10.3390/ijms21113990

**Published:** 2020-06-02

**Authors:** Juliana H. Azambuja, Nils Ludwig, Saigopalakrishna S. Yerneni, Elizandra Braganhol, Theresa L. Whiteside

**Affiliations:** 1Department of Pathology, University of Pittsburgh School of Medicine, Pittsburgh, PA 15213, USA; julianahazambuja@hotmail.com (J.H.A.); ludwign@upmc.edu (N.L.); 2UPMC Hillman Cancer Center, Pittsburgh, PA 15213, USA; 3Programa de Pós-Graduação em Biociências, Universidade Federal de Ciências da Saúde de Porto Alegre (UFCSPA), Porto Alegre 90050-170, Brazil; elizbraganhol@yahoo.com.br; 4Department of Biomedical Engineering, Carnegie Mellon University, Pittsburgh, PA 15213, USA; syerneni@andrew.cmu.edu; 5Departments of Immunology and Otolaryngology, University of Pittsburgh School of Medicine, Pittsburgh, PA 15213, USA

**Keywords:** glioblastoma, glioblastoma--derived exosomes (GBex), tumor-associated macrophages (TAMs), macrophage reprogramming, TAM-derived exosomes, Arginase-1

## Abstract

Interactions between tumor cells and tumor-associated macrophages (TAMs) are critical for glioblastoma progression. The TAMs represent up to 30% of the glioblastoma mass. The role of TAMs in tumor progression and in the mechanisms underlying tumor growth remain unclear. Using an in vitro model resembling the crosstalk between macrophages and glioblastoma cells, we show that glioblastoma-derived exosomes (GBex) reprogram M1 (mediate pro-inflammatory function) and M2 (mediate anti-inflammatory function) macrophages, converting M1 into TAMs and augmenting pro-tumor functions of M2 macrophages. In turn, these GBex-reprogrammed TAMs, produce exosomes decorated by immunosuppressive and tumor-growth promoting proteins. TAM-derived exosomes disseminate these proteins in the tumor microenvironment (TME) promoting tumor cell migration and proliferation. Mechanisms underlying the promotion of glioblastoma growth involved Arginase-1+ exosomes produced by the reprogrammed TAMs. A selective Arginase-1 inhibitor, nor-NOHA reversed growth-promoting effects of Arginase-1 carried by TAM-derived exosomes. The data suggest that GBex-reprogrammed Arginase-1+ TAMs emerge as a major source of exosomes promoting tumor growth and as a potential therapeutic target in glioblastoma.

## 1. Introduction

Glioblastoma, also called grade IV glioma, represents about 50% of all brain neoplasms [1]. The five-year survival rate of this lethal and aggressive tumor has improved only marginally in the last decade, with only 10% of patients surviving more than five years. There is an unmet and urgent need for new targets and novel therapies for glioblastoma [2,3]. Interactions between tumor cells and tumor-infiltrating macrophages have been reported to be important for disease outcome in several types of cancer, including glioblastoma [4]. Brain tumors effectively recruit immune cells derived from the blood, up to 30% of the glioblastoma mass consists of macrophages, suggesting that that these cells are involved in the promotion of tumor progression and in chemoresistance [5,6].

Cellular plasticity has been shown to be a characteristic feature of macrophages and microglia [7]. Macrophages are classified as M1 (mediate pro-inflammatory functions) and M2 (mediate anti-inflammatory functions) [8]. In the tumor microenvironment (TME), macrophages are known to participate in numerous cellular functions, including cell proliferation, migration, angiogenesis, and metastasis [9]. Nevertheless, mechanisms essential for the education of naïve macrophages to tumor-associated macrophages (TAMs) with tumor-promoting functions remain unclear. 

A number of immunoregulatory mechanisms have been identified that are responsible for establishing an immunosuppressive TME [10]. These mechanisms include downregulation of tumor-associated antigens (TAA) expression, activation of immunosuppressive cells, such as regulatory T cells (Treg) and M2-like macrophages, production of immunosuppressive cytokines, alteration of glycolysis and induction of oxidative stress [5,10,11]. In a recent study, we reported that glioblastoma-derived exosomes (GBex), suppressed CD69 and NK2GD expression in T and NK cells, respectively, inhibited CD8+ T cell proliferation and cytokine production and induced the M2-like phenotype in macrophages [12]. In this study, we demonstrate that GBex play a key role in reprogramming functions of various macrophage subsets present in the TME. We show that GBex can reprogram not only naïve or M1 macrophages promoting their differentiation to M2 macrophages, but they can also educate M2 macrophages, converting them into strongly immunosuppressive TAMs. Further, we demonstrate that GBex-educated and reprogrammed macrophages produce exosomes which are responsible for tumor-promotion.

## 2. Results

### 2.1. Glioblastoma Cell-Derived Exosomes (GBex) Reprogram Macrophages 

To investigate the role of glioblastoma cells in reprogramming the phenotype and functions of macrophages, various human macrophage subsets (naïve, classically activated, or alternatively activated macrophages) were co-incubated with GBex. To convert naïve to M1 macrophages, primary human macrophages were pretreated with Lipopolysaccharide; conversion to M1 was confirmed by expression of the M1 macrophage phenotypic markers, CD80, CD86, major histocompatibility complex) class II (HLA-DR), and interferon (INF)-γ. To convert naïve to M2 macrophages, treatment with IL-10 was performed, and the conversion was ascertained by measuring expression of arginase-1, IL-10, and CD206 in M2 cells. Co-incubation of classically activated macrophages with GBex invariably resulted in significantly increased expression of arginase-1, IL-10, and CD206 (i.e., the M2-like phenotype) and in decreased expression of M1-like markers (CD80, CD86, HLA-DR, and INF-γ). Because tumors can modulate immune cells to acquire a tumor-associated immunosuppressive phenotype, we also determined the expression of immunomodulatory markers including programmed death ligand-1(PDL-1), Fas ligand (FasL), and Cytotoxic T lymphocyte antigen4- (CTLA-4) on the GBex-treated macrophages. As shown in Figure 1 and Appendix A expression levels of immunomodulatory proteins CTLA-4 and PDL-1 were significantly increased in macrophages treated with GBex compared to controls. Taken together, these results demonstrated that GBex reprogrammed the different macrophage subsets, resulting in the acquisition of the TAM phenotype which bears similarities with that of M2-like macrophages. Importantly, GBex did not induce death in macrophages (Appendix A). To differentiate these GBex-reprogrammed cells from M2-like macrophages, we refer to them as tumor-associated macrophages or TAMs.

### 2.2. Isolation and Characterization of TAM-Derived Exosomes

We collected supernatants from cultures of TAMs, i.e., GBex-reprogrammed macrophage subsets (M0, M1, and M2), and used them for isolation of TAM-derived exosomes by size exclusion chromatography (SEC). These supernatants contained from 20 to 40 μg exosome protein/mL. M1-TAMs produced significantly higher levels of total exosome proteins (35.7 μg/mL) relative to M1 macrophages (18.3 μg/mL) (Figure 2A,B). Transmission electron microscopy showed that vesicles isolated from supernatants of TAMs were increased in size and appeared more homogeneous (Figure 2A). The presence of Tumor susceptibility gene 101 (TSG101) and CD9 in the vesicle cargo was shown by Western blots, confirming their origin from the endocytic compartment of the parent cells and placing them in the category of small exosomes (Figure 2C). 

Based on the notion that exosomes carry a molecular cargo which is partly similar to that of their parent cells, we compared protein profiles on the surface of GBex-treated macrophages with those of exosomes produced by these macrophages. Western blots in Figure 3A show that the TAM-derived exosomes carried PDL-1 and arginase-1, known to mediate immunosuppression and tumor progression, and that the protein profiles of these exosomes were qualitatively and quantitatively similar to those of parent macrophages (Figure 3B). However, the differences founding the exosomes cargo between the Western blot and the flow cytometry experiment, cloud be explained because in the Western blot (Panel A) we are looking at markers present both in the lumen and on the surface of total exosomes, but in the flow cytometry were are looking for the surface content in CD63 captured exosomes. At the end, this initial profiling of TAM-derived exosomes showed that their molecular content was similar to that of the GBex reprogrammed parent cells. Thus, these TAM-derived exosomes might be expected to also mediate immunosuppressive and pro-tumor functions.

### 2.3. TAM-Derived Exosomes Show Pro-Tumor Activities

Pro-tumor functions of TAM-derived exosomes were evaluated in transwell migration assays. Promotion of glioblastoma cell migration was evident when exosomes produced by M2-like macrophages were used (Figure 4A). Furthermore, exosomes produced by M1 macrophages decreased tumor cell migration by 60% relative to the exosomes from M2 macrophages, which mediated strong tumor cell migration. When macrophages were reprogrammed to TAMs after GBex treatment, they produced exosomes which enhanced glioblastoma cell migration independently of the initial phenotype status of macrophages. Thus, glioblastoma cell migration was enhanced by 900% using exosomes from M0 TAMs, 300% with exosomes from M1 TAMs, and 1000% with exosomes from M2 TAMs (Figure 4B). 

Exosomes released from various macrophage subsets treated with GBex also promoted glioblastoma cell proliferation. As expected, M2-derived exosomes enhanced glioblastoma cell numbers and bromodeoxyuridine (BrdU) incorporation to the greatest extent (Figure 5A–C). Exosomes released from TAMs also induced alterations in the cell cycle of glioblastoma cells, enhancing the numbers of cells in the synthesis phase (Figure 5B). In addition, all subsets of macrophages co-incubated with GBex (i.e., all TAM subsets) produced exosomes enhancing tumor cell proliferation (M0-TAMs: 200%, M1-TAMs: 160%, and M2-TAMs: 300%) (Figure 5D). Important to note that BrdU is a short-term experiment for 4 hours, whereas the proliferation assay was performed done over a period of 4 days. 

We also evaluated the impact of the exosomes released from macrophage subsets in response to the chemotherapy. Temozolomide (TMZ) decreased 60% glioma proliferation. However, in the group with glioblastoma cells co-incubated with TAM-derived exosomes, we observed an increase in cell numbers after TMZ treatment suggesting protection against TMZ cytoxicity perhaps because treatment with the TAM-derived exosomes enhanced glioblastoma proliferation (Appendix A). 

Taken together, these results demonstrated that exosomes produced by TAMs showed pro-tumor activities by enhancing glioblastoma cell proliferation and migration.

### 2.4. Arginase-1 Activity of TAM-Derived Exosomes is Inhibited by Nor-NOHA

Arginase-1, an enzyme responsible for conversion of arginine to ornithine and urea, is found on the macrophage surface (Figure 1A). We observed that Arginase-1 expression was increased in exosomes isolated from supernatants of TAMs (Figure 3A,B). To determine whether the increase of arginase-1 expression in TAM-derived exosomes is responsible for the promotion of glioblastoma cell proliferation, we incubated tumor cells in the presence of arginine or the arginase inhibitor (nor-NOHA) for 24 h and measured cell viability. As shown in Appendix A, arginine, when added in excess, suppressed the viability and thus growth of glioblastoma cells. Similar growth inhibitory effects were mediated by the arginase-1 inhibitor, confirming the key role of the arginase pathway in glioblastoma growth. We also showed the arginase inhibitor blocked the ability of TAMs-derived exosomes to induce glioblastoma proliferation (Figure 6). As to the inhibitor (Nor-NOHA), it has the same effect on all groups of exosomes, irrespective their origin. This is indeed expected, because the exosomes from macrophages (no matter the source) always carry arginase, but at different levels (Figure 3B). This data suggest that glioblastoma growth induced by TAMs-derived exosomes was in part mediated by arginase-1 expression/activity. 

## 3. Discussion

In this study, we reported that glioblastoma cells induced a pro-tumor phenotype which bears similarities with that of M2-like macrophages in different subsets of macrophages via exosome-mediated communication. GBex-activated tumor-associated macrophages, in turn, responded by releasing their own exosomes which promoted glioblastoma cell migration and proliferation in vitro. The promotion of tumor growth was in part mediated by arginase-1 carried by TAM-derived exosomes. Our findings reveal a novel mechanism for macrophage-glioblastoma crosstalk in the TME and provide additional evidence for the important role exosomes play in reprogramming the TME and regulating immune cell-tumor cell communication.

Inflammation in the TME is a hallmark of cancer progression [13]. In this context, macrophages accumulating in the tumor stroma have a dual influence on the occurrence and development of cancers based on the macrophage activation status. Thus, while anti-tumor effects are mediated by classically activated (M1) macrophages, pro-tumor activities are driven by alternatively activated (M2) macrophages [14,15]. We have previously shown that GBex present in glioblastoma cell conditioned media can redirect macrophages to acquire the pro-tumor M2-like phenotype [6,16]. These M2 macrophages go on to produce their own exosomes, which contribute to establishing a pro-tumor milieu. Indeed, emerging evidence emphasizes an important role for stromal cell-derived exosomes as mediators of cell-to-cell communication within the TME [17]. This study was designed to investigate mechanisms used by the GBex-reprogrammed macrophages to maintain the pro-tumor microenvironment in glioblastoma. Our results show that exosomes produced by TAMs significantly contributed to tumor progression. 

Exosomes produced in the TME are derived from a variety of different cells and presumably play different regulatory roles in the tumor growth. While much attention has been devoted to documenting GBex-mediated reprogramming of various stromal cells, including macrophages, little is known about the role in tumor cell progression of exosomes these reprogrammed cells produce and release in the TME. As suggested above, macrophage-derived exosomes have a dualistic role in preventing or supporting tumor progression. Studies have suggested that macrophage-derived exosomes can inhibit metastasis of ovarian tumor cells or cause fibrosarcoma regression [18,19]. Others reported that exosomes secreted by a macrophage cell line supported aggressiveness of pancreatic ductal adenocarcinoma as well as breast and gastric cancers [20,21,22]. None of these earlier studies considered a possibility that GBex-reprogrammed macrophages secreted exosomes which contributed to establishing the pro-tumor milieu. 

Our previous work showed that macrophages stimulated by GBex secreted soluble factors promoting cancer progression [12]. Here, we have extended our results to show that these soluble factors include exosomes which are produced and released by macrophages reprogrammed by GBex in the TME. It is the exosomes produced by these TAMs that are responsible for enhancing tumor progression. Thus, GBex once released by glioblastoma cells, initiate a cascade of reprogramming promoting tumor growth and exerting immunosuppressive effects in various subsets of macrophages in the TME. Thus, GBex-educated TAMs emerge as a major source of exosomes that drive tumor growth in glioblastoma. 

Functionally, exosomes can transfer a variety of proteins, DNA, and RNA, which might have a broad variety of effects on cancer progression. In this study, the protein that was significantly enriched in GBex-reprogrammed macrophages and in exosomes they produced was arginase-1. The enzyme arginase (ARG, EC 3.5.3.1) hydrolyzes L-arginine to the products L-ornithine and urea [23]. The impact of L-arginine metabolism disorders on both carcinogenesis and on antitumor immune system activity has recently received a lot of attention [24]. Specifically, Czystowska-Kuzmicz and co-workers showed that arginase+ exosomes derived from an arginase+ ovarian cell line impaired functions of human and murine T-cells by blocking their proliferation and reducing expression levels of the CD3ζ and CD3ε chains [25]. To the best of our knowledge, we are the first to report arginase-1 expression on exosomes produced by GBex-reprogrammed TAMs. Several previous studies indicated that blocking arginase activity may be an attractive therapeutic target to promote anticancer effects [23,24]. Here, we show the potential of arginase inhibition for blocking TAM-mediated pro tumor activity. Although we did not do a global analysis looking at the whole cargo content of TAM-derived exosomes, we suspect that TAM-derived exosomes may carry other agents that influence the migration, resistance, invasion, and other biologic functions of glioma cells. Further studies are needed to understand the whole role of TAMS-derived exosomes (from macrophages and also microglia) in the complex tumor biology and microenvironment.

## 4. Materials and Methods 

### 4.1. Culture of Glioblastoma Cells

The glioblastoma cell line U251, was obtained from American Type Culture Collection (ATCC). Cells were grown in DMEM (Lonza Inc.) supplemented with 10% (*v/v*) exosome-depleted and heat-inactivated fetal bovine serum (FBS) (Gibco, Thermo Fisher Scientific) at 37 °C and in the atmosphere of 5% CO_2_ in air. For exosome isolation, 4 × 10^6^ U251 cells were cultured in 150cm^2^ cell culture flasks containing 25 mL cell culture medium. After 72h, supernatants were collected and used for the isolation of glioblastoma cell-derived exosomes (GBex). The choice of cell line was based on an article and previous data that showed no difference in the immunoregulatory effect between glioblastoma cell lines [12].

### 4.2. Macrophages

Blood samples were obtained from healthy donors, and peripheral blood mononuclear cells (PBMCs) were isolated by Ficoll® Paque Plus (GE Healthcare Bioscience) by centrifugation as previously described [26]. All subjects donating blood specimens signed an informed consent approved by the Institutional Review Board of the University of Pittsburgh (IRB #960279 and IRB #0506140). 

Monocytes were separated by adherence to plastic. For the differentiation into macrophages, isolated monocytes were cultured in the presence of Granulocyte-macrophage colony-stimulating factor (GM-CSF) (50 ng/ml, Peprotech) in Roswell Park Memorial Institute Medium (RPMI) supplemented with 10% (*v/v*) exosome-depleted and heat-inactivated fetal bovine serum (FBS) for 7 d as previously described [12].

Macrophages were separated into 3 different groups: (i) M0, Naïve macrophages maintained in RPMI medium; (ii) M1, classically activated macrophages maintained in Lipopolysaccharide (LPS) (10 ng/mL); and (iii) M2, alternatively activated macrophages maintained in Interleukin 10 (IL-10) (10 ng/mL). The macrophage polarization was performed for 6h. Next, M0, M1, or M2 macrophages were co-incubated with GBex (25 µg protein/mL) for additional 12 h. The same volume of PBS was added to control wells. After 12 h of co-incubation with GBex, the medium was replaced with fresh complete medium and after 48h, the supernatant was collected for exosome isolation. 

### 4.3. Exosome Isolation 

Exosomes were isolated from culture supernatants as previously described [26]. Briefly, supernatants were centrifuged at 2000× *g* at room temperature (RT) for 10 min and then at 10,000× *g* for 30 min at 4 °C. They were then filtered using 0.22 μm syringe-filters (Millipore, Burlington, MA, USA). Supernatants were concentrated to 1 mL on Vivacell 20 filter units (MWCO 100,000, Sartorius Corp, Bohemia, NY, USA). Aliquots (1 mL) of concentrated supernatants were loaded on mini-SEC columns (18), and exosomes were eluted with PBS (pH 7.4). Exosomes were collected in fraction #4 (1 mL). For some experiments, #4 mini-SEC fractions were concentrated (1 µg/µL) using 100,000 Molecular weight cut-off Vivaspin 500 Centrifugal Concentrators (Sartorius Corp, Bohemia, NY, USA) at 5000× *g* for 5–10 min.

### 4.4. Characterization of Isolated EVs

Exosomes were characterized as previously described [26]. Their morphology and size were evaluated by transmission electron microscopy (TEM). The particle concentration and size distribution were determined by tunable resistive pulse sensor (TRPS) technology using a qNano instrument (Izon Science). The exosome protein concentration was measured by BCA (Thermo Scientific, Rockford, lL, USA, REF 23225). Protein cargos of exosomes were analyzed by Western blotting and flow cytometry as previously described by us [26]. Antibodies used for immunodetection of the exosome cargos are listed in Appendix A.

### 4.5. Flow Cytometry

Cells were suspended in the staining buffer (PBS + 3% BSA). Fluorochrome-conjugated antibodies used for staining are listed in Appendix A. For intracellular staining, cells were first fixed with the eBioscience™ IC Fixation Buffer (00-8222-49), permeabilized with eBioscience™ Permeabilization Buffer (00-8333-56) and stained. For cell surface staining, cells were blocked with anti-FcR reagent and then incubated for 30 min at RT with fluorochrome-labeled antibodies at dilutions determined by pre-titrations, including appropriate isotype controls. After washing in flow citometry buffer, cells were immediately analyzed using an Accuri flow cytometer (BD Bioscience).

### 4.6. Annexin V-Based Apoptosis Assays

Apoptosis was measured using FITC Annexin V Apoptosis Detection Kit (556547, BD Bioscience) according to the manufacturer’s instructions with Accuri flow cytometer (BD Bioscience, San Jose, CA, USA).

### 4.7. Migration Assay 

Glioblastoma cell migration was evaluated using a 24-well transwell chamber (Corning Inc.) as previously described [27]. For this assay, 2.5 × 10^4^ U251 glioma cells were seeded in the upper chamber with an 8-μm pore-size insert in DMEM only and allowed to migrate towards a serum-free medium + exosomes (10 µg) placed in the lower chamber. After 24 h, non-migrating cells in the upper chamber were removed with a cotton swab. and the remaining cells were fixed in methanol for 15 min. Cells that migrated to the lower surface of the membrane were stained with 0.5% crystal violet diluted in 20% (*v/v*) methanol. The number of migrated cells was counted in a light microscope in six randomly selected regions of interest at 20× magnification using an Olympus BX51 microscope (Olympus America). Cell quantification was performed by processing all obtained images using ImageJ software (http://imagej.nih.gov/ij/). 

### 4.8. Proliferation Assay

Glioblastoma cells were seeded in wells of a 24-well plate (2 × 10^3^ cells in 500 µL). After 2 h, 10 µg of exosomes isolated from supernatants of macrophages or glioblastoma cells were added. Following a 4 d incubation at 37 °C in 5% CO_2_, glioblastoma cells were harvested, and their cell number was determined by flow cytometry (Accuri, BD Biosciences). For the Temozolomide sensitivity experiments, cells were also incubated with 200 µM of Temozolomide (TMZ, T2577, Sigma) together with exosomes, and the cell number and the apoptotic index were determined as previously described. In some experiments, 500 nM of N-hydroxy-L-arginine (nor-NOHA; Cayman Chemical), an arginase inhibitor, was added to co-cultures.

### 4.9. BrdU Assay

Glioblastoma cells were seeded in wells of a 24-well plate (2 × 10^3^ cells in 500 µL). After 2 h, 10 µg of exosomes isolated from macrophages or glioblastoma cells were added, as previously described. After incubation for 6h at 37 °C in 5% CO_2_, glioblastoma cells were incubated with 10μM bromodeoxyuridine (BrdU) for additional 2 h. Then, cells were incubated with anti-BrdU-FITC antibody according to the manufacturer’s recommendations (559619, BD Biosciences), and the percentage of BrdU incorporating cells was analyzed by flow cytometry (Accuri; BD Biosciences) [6].

### 4.10. Cell Cycle 

Glioblastoma cells were seeded and cultured as described above. After co-cultures, cells were fixed, permeabilized and their DNA content was analyzed as previously described [27].

### 4.11. MTS Assay

U251 GB cells were seeded in wells of 96-well plates (5 × 10^3^ cells/well) overnight. Cells were co-incubated with Arginine (10 µM, A5006, Sigma) or an arginase inhibitor (nor-NOHA, 10 µM, 1140844-63-8, Cayman Chemical) for 24 h. The MTS (3-(4,5-dimethylthiazol-2-yl)-5-(3-carboxymethoxyphenyl)-2-(4-sulfophenyl)-2H-tetrazolium) cell viability assay was performed according to the manufacturer’s instructions (ab197010, Abcam), and cell viability was calculated using Prism 7.0 software (Prism GraphPad Software) according to the following formula: cell viability rate (%) = (OD490 of treated cells/OD490 of control) × 100%. 

### 4.12. Statistical Analysis

Data were analyzed using GraphPad Prism (v8.0). Results are expressed as means ± SEM (standard error of the mean). Differences between groups were assessed by a one-way ANOVA (analysis of variance) and were considered significant at *p* < 0.05.

## 5. Conclusions

In conclusion, the in vitro model of the glioblastoma microenvironment we present appears to be useful for providing new insights into the mechanisms by which macrophages that accumulate in the tumor are regulated. We demonstrate that glioblastoma cell-derived exosomes, GBex, induce pro-tumor activation of macrophages, which upon conversion into TAMs release an abundance of immunosuppressive and tumor growth-promoting exosomes. It is these “secondary” exosomes that may be the major mechanism contributing to tumor growth. We also provide the first evidence for the role of arginase-1 in the glioblastoma progression mediated by TAM-derived exosomes. Hereby, we identify arginase-1+ TAMs as a potential therapeutic target in glioblastoma. This knowledge might permit the development of new strategies for targeting tumor growth-promoting macrophages in the TMA.

## Figures and Tables

**Figure 1 ijms-21-03990-f001:**
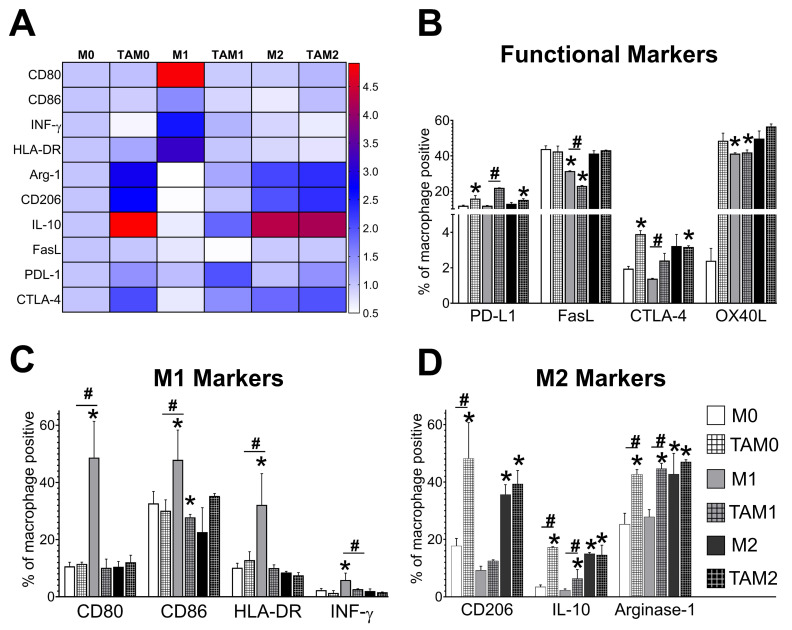
GBex induced a phenotype like TAMs in different macrophages subsets. (**A**) The heat map shows the fold of decrease or increase in expression of macrophage markers evaluated by flow cytometry and considering M0 as control (**B**) percent of macrophages positive for functional immunomodulatory markers PDL-1, FasL, CTLA-4 and OX40L); (**C**) M1 markers (CD86, CD80, HLA-DR INF)-γ; (**D**) M2 markers (CD206, Arginase-2 and IL-10). Data were analyzed by flow cytometry as described in Materials and Methods. Data represent mean ± SD of three independent experiments performed in triplicate. Data were analyzed by ANOVA followed by Tukey post hoc. *Significantly different from macrophage M0 cells (white bar); #significantly different between macrophage and TAM cells (*p* < 0.05).

**Figure 2 ijms-21-03990-f002:**
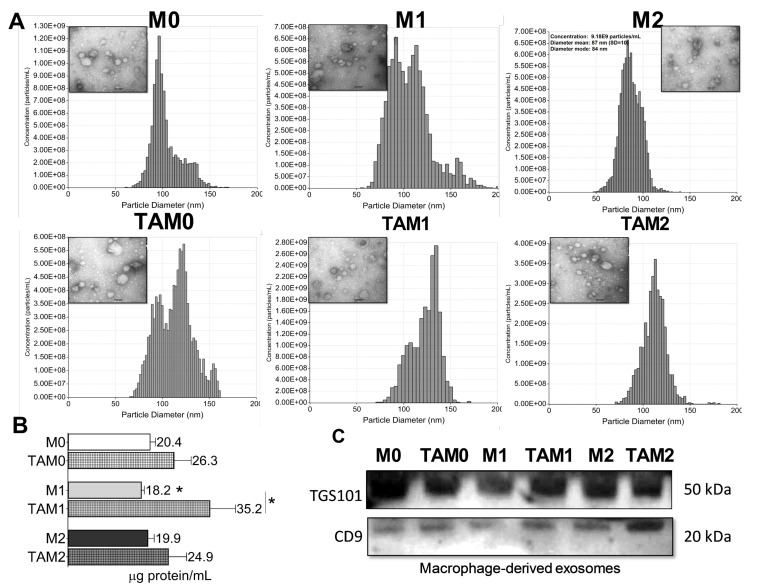
Characteristics of exosomes produced by macrophages or GBex-reprogrammed TAMs. (**A**) Results of qNANO analyses and representative transmission electron microscopy (TEM) images providing concentrations and sizes of exosomes produced by macrophages or TAMs; (**B**) Total protein levels isolated from supernatants of macrophages or TAMs. The data are mean values ± standard error (SEM) from 3 experiments Data were analyzed by ANOVA followed by Tukey post hoc. *Significantly different from control cells at *p* < 0.05; (**C**) Western blot profiles of exosomes produced by macrophages or TAMs. Each lane was loaded with 10 μg exosome protein. Note the presence of exosome markers CD9 and TSG101.

**Figure 3 ijms-21-03990-f003:**
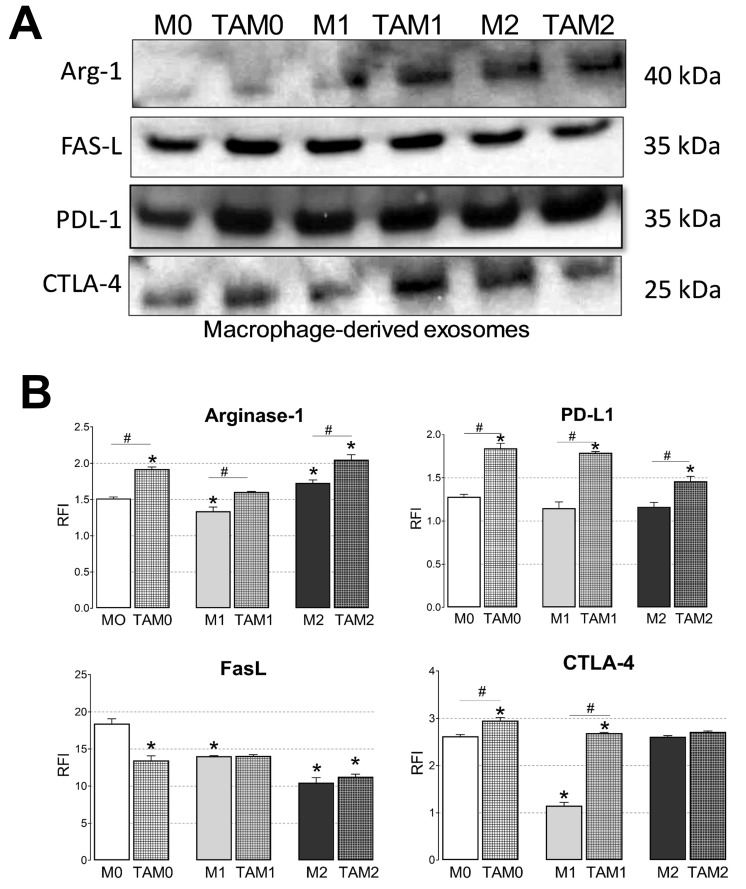
Immunosuppressive cargos of exosomes produced by macrophages or TAMs. (**A**) Representative Western blots of exosomes isolated from macrophages or TAMs. Equal amounts of exosomal protein (10 μg) were loaded per lane; (**B**) Flow cytometry results for the detection of PDL-1, FasL, CTLA-4, and Arginase-1 carried on exosomes produced by macrophages or TAMs. Exosomes were immunocaptured with anti-CD63 mAb for on-bead flow cytometry as described in Materials and Methods. Data are relative fluorescence intensity (RFI) values ± SEM from three independent experiments Data were analyzed by ANOVA followed by Tukey post hoc. *Significantly different from the control at *p* < 0.05 and # Significant difference between macrophages and TAMs at *p* < 0.05.

**Figure 4 ijms-21-03990-f004:**
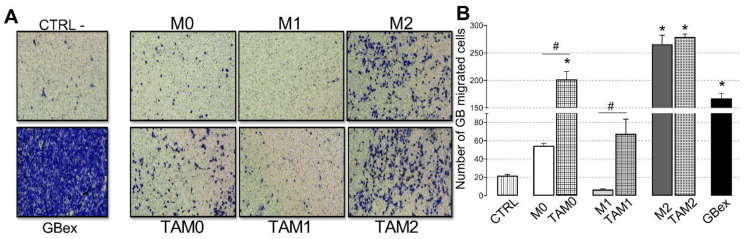
Exosomes produced by TAMs enhance glioma cell migration. U251 glioma cells were suspended in Dulbecco′s Modified Eagle′s - Medium (DMEM) and were seeded in the upper chamber of transwell units. Exosomes (10 µg) produced by macrophages or were added to the lower transwell chamber. (**A**) Representative light microscopy images of crystal violet-stained glioblastoma cells accumulating on the lower surface of the membrane; (**B**) Numbers of glioblastoma cells migrating to the lower chamber. Data are mean values ± SEM from three independent experiments. Data were analyzed by ANOVA followed by post hoc comparisons (Tukey–Kramer test). *Significantly different from the control at *p* < 0.05 and # Significant difference between macrophages and TAMs at *p* < 0.05.

**Figure 5 ijms-21-03990-f005:**
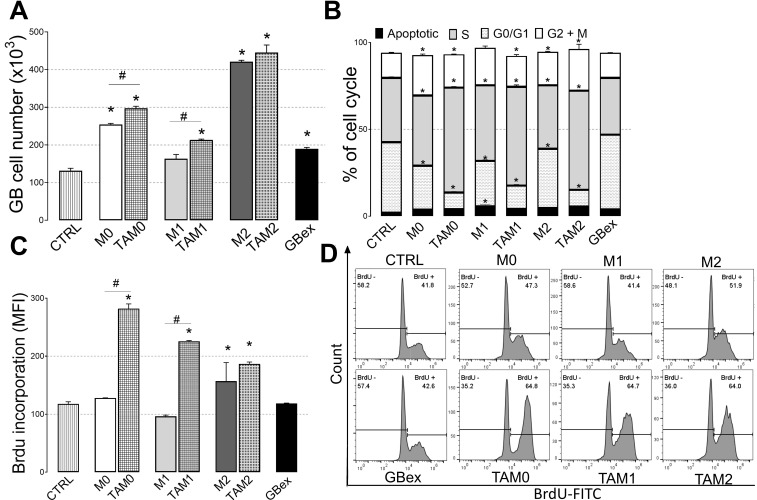
Exosomes produced by TAMs enhance glioma cell proliferation. Glioma cells were incubated in the presence of 10µg exosomes produced by different subsets of macrophages or TAMs. Control cells were incubated in DMEM. (**A**) Cell proliferation was assessed by counting cells in a flow cytometer; (**B**) Flow cytometry based determination of the cell cycle distribution (Apoptotic, G1/G0, S, and G2/M) in U251 cells; (**C**) Bromodeoxyuridine (BrdU) incorporation by glioblastoma cells. Combined data are mean values ± SEM from three independent experiments; (**D**). Representative flow cytometry results from one experiment showing BrdU incorporation by glioblastoma cells incubated with exosomes produced by macrophages or TAMs. Results were analyzed by one-way ANOVA, followed by Tukey’s multiple comparisons test. *Significantly different from the control at *p* < 0.05 and # Significant difference between macrophages and TAMs at *p* < 0.05.

**Figure 6 ijms-21-03990-f006:**
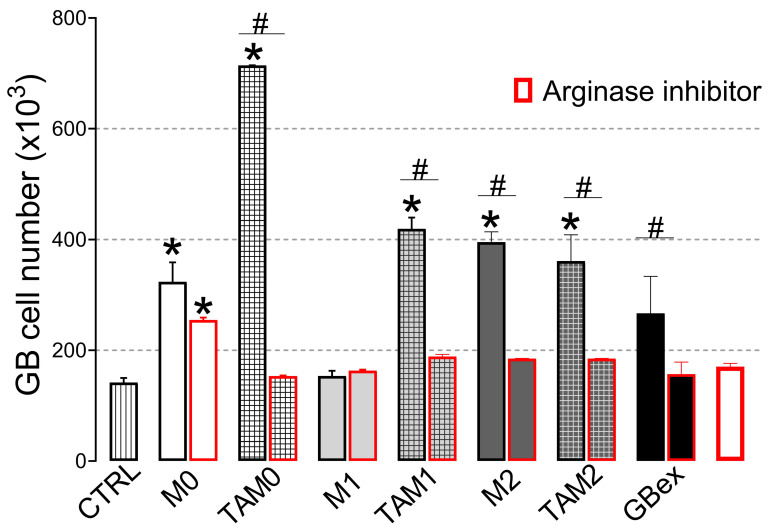
Arginase-1+ exosomes released from TAMs mediate glioblastoma cell proliferation in vitro. Glioblastoma cells were incubated with 10 µg of exosomes produced by of macrophages or TAMs in the presence or absence of an arginase inhibitor (N-hydroxy-L-arginine) for 4 days. Control cells were exposed to DMEM. Cell proliferation was assessed by counting cells. Data are mean cell counts ± SEM from three independent experiments. Results were analyzed by one-way ANOVA, followed by Tukey’s multiple comparisons test. *Significantly different from the control at *p* < 0.05 and # Significant difference between macrophages and TAMs at *p* < 0.05.

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
