# Peer review of "Arginase-1+ Exosomes from Reprogrammed Macrophages Promote Glioblastoma Progression"

_ijms, 2020, doi:10.3390/ijms21113990_

Round 1

Reviewer 1 Report

In this manuscript, authors have evaluated the impact of glioblastoma cell derived exosomes on tumor promoting macrophages (such as TAM).  Briefly, they have shown that tumor derived exosomes can convert tumor promoting TAM from M1or M2 macrophages, which upon conversion TAM can produce immunosuppressive but potent tumor growth stimulating exosomes that supports glioma growth. Although this is an interesting study and identify a novel mechanism of tumor exploitation of the immune effector cells in the tumor microenvironment, several issues need to be addressed before the manuscript is accepted for publication.

Glioblastoma patient derived tumors are extremely heterogeneous, having divergent genetic backgrounds with different methylation status. Most of the in vitro experiments in this study were performed using only one glioblastoma cell line (U251).  Thus, main conclusion from this finding cannot be ascertained based on only one cell line tested.  Some of the key experiments need to be repeated using at least one more glioma cell line which is genetically different from U251.

Figure 1: Authors need to provide the clear statistics for panel B,C and D; such as sample size (n) for each group and P values (at least in the figure legend).

Fig. 1A: What method authors used for generating the heat map to show the  fold of decrease or increase of macrophage markers?

Fig. 1B: Why did the authors select OX40L as functional immunosuppressive marker for macrophages?  Authors did not mention the documentation of this marker even in the figure legend.

Fig 2A: Quality of figures for panel A is of poor quality.  Hard to read “y-axis, or the images adjacent to each data set. Please improve the quality.

Fig 3: Flow cytometry data (panel B) do not reflect the Western blots data for all 4 markers tested for exosomes isolated from or produced by macrophages.  Authors need to explain these apparent discrepancies.

Fig 4: Authors have shown that exosomes produced by TAMs enhance glioma cell migration. Based on Figure 3, the exosomes isolated from macrophages TAM1 and TAM2 have almost equal amount of Arg-1, FAS-L or PD-L1. Why the migration of TAM1 and TAM2 are then markedly different?  Please explain or discuss.

Have the authors tested effect of TAM derived exosomes on microglia cells, the resident cell of the brain?

Minor comments:

Abbreviations for some markers are not consistent throughout the manuscript, in figures and text (for example IFN- γ, FAS-L, PD-L1 and so on).  Please check throughout the manuscript and use the correct abbreviations and be consistent.  CTLA-4 should be Cytotoxic T lymphocyte antigen-4 not 1, please correct.

 Descriptions of methods in figures legends (for example for Figures 1,2 and 4) should all be moved to Materials and Methods.  

Please improve the figure quality as best as possible.

Statistics for each figure must be mentioned clearly as and where they are missing.

Reviewer 2 Report

Juliana H. Azambuja and collaborators show an in vitro study based on the pro-tumourigenic properties of GB- and TAM-associated exosomes. They use a panel of M1- and M2-like biomarkers to characterise the features of the exosomes and their impact on tumour biology. The main novelty of the study lies in the description of a subtype of exosomes with Arg-1 over-expression with strong tumour-supporting properties. The use of Arginine-depleting drugs or Arg-1 inhibitors have already brought important attention in the field as potential anti-cancer therapies.  

In general, there are a few concerns regarding the interpretation of their findings. The statistical approach and the way their results are analysed seem wrong on some occasions. Furthermore, the authors claim that they educate M2 macrophages converting them into strongly immunosuppressive TAMs. The conjunction of the result does not refute such statement. For most of the results, M2 and TAM2 groups don´t seem to shed significant differences in any of the analysed parameters (except just for Arg1 and PD-L1), therefore, this reviewer finds hard to understand the authors´ statement.   

It would also have been convenient to have access to their recent publication at Neuro-Oncology advances in order to fully comprehend the impact of their research and findings.

I would like to expose my concerns regarding the way authors represent their findings as follows:

With regard to this manuscript, the authors mention the use of an in vitro model resembling the TME. It was hard to find any info regarding the features or components of such model. The tumour microenvironment is formed by several different cell types (immune cells and brain resident cells). As far as this reviewer could understand, there is no mention to any other cell type apart from human-derived macrophages and U251. Therefore, the use of the term TME is wrong.

Additionally, the cross-talk between GB cells and TAMs is not exclusive for macrophages. Microglia is one of the main components of TAMs (or GAMs in this case –Glioma-associated macrophages). Despite understanding the limitations to distinguish them in an in vivo context, since they use an in vitro approach, it is essential to study microglia-associated exosomes to legitimately use the term TAM-associated exosomes.

In Figure 1, the authors fail to explain the meaning of the * and # symbols. The heat map seems to lead to an over-estimation of the results. For instance, in the case of Arg-1 expression in TAM2, it seems to fall between 2.5 and 3 time-fold increased, whilst on the graph (1D) is barely 2-fold increased. The heat map has to be thoroughly revised or substituted by a table with the results expressed with mean and SEM. Actually, the table should be included in any case.

In Figure 2, the resolution of the figure is poor and the labels can be barely seen, which complicated the revision of the data. The authors show the average of protein/mL from supernatants. It was hard to check whether those levels had any relation with the number of exosomes released. It would be interesting to analyse whether the exosomes released by macrophages or TAMs change the content of proteins within.

Again, a table showing the number of exosomes released by all subtypes of experimental cells would help the readers to understand the results. Once this is done, the statistical analysis should asses differences between all groups.

In Figure 3, the western blot is poor. It is hard to check whether levels of Arg-1 in M1 is lower owing to an artefact or real downregulation. Authors should provide a better image, otherwise, the results seem wrong. In the same vein, the expression of CTLA-4 between M2 and TAM2 seem extremely different but in the graphs show similar values.

In Figure 4, again pictures do not seem to fully correlate with their findings. For instance, the number of cells in TAM0 and TAM2 look extremely different. However, in the graphs there seem to be just 60 or 70 cells difference.

In Figure 5, M2 and TAM2 experimental groups seem to have big differences in the number of cells in S phase whilst no differences in BrdU are appreciated. Please elaborate.

Lastly, in Figure 6, there is a paragraph above the figure that should be deleted.

The numbers of GB cells are significantly different as those shown in Figure 5A. Since the design of the experiment seems to be the same, it is very contradictory the massive differences between groups that shall have undergone the same procedure. For instance, the number of GB cells in the TAM0 group is 300 in Figure 5A but around 700 in Figure 6. That seems to help authors to achieve the provided statistical differences. It would be convenient either to normalise all results or to throw a proper explanation for such differences. The same for the TAM1 and TAM2 groups.

In this particular experiment, it is important to compare the effect of Arg1 inhibitor amongst all groups. It seems that Nor-NOHA has the same effect on any type of exosomes irrespective their origin.

Round 2

Reviewer 2 Report

Juliana H. Azambuja and collaborators have significantly improved their manuscript. They have also addressed all comments, although, I would like to ask for further explanation in a couple of minor points:

Many of the properties of M2-like macrophages are also characteristic of tumor-associated macrophages (TAMs). And the results exposed by the authors refute such a statement. As previously mentioned, most of the experiments in this manuscript do not show differences between M2 and TAM2 macrophages. Therefore, GBex did not turn M2 macrophages into TAMs because they already possessed pro-tumoural features. Therefore, the comment in line 19 : “we show that glioblastoma-derived exosomes (GBex) reprogram M1 and M2 macrophages, converting them into TAMs”… seem an overestimation of their findings.

Finally, in the same vein, the title for paragraph “2-4 TAM-derived exosomes carry arginase-1” is biased. M2 macrophages also seem to carry Arg1 and to be affected by Nor-NOHA. Therefore, the title should point at the effect of the Arginase inhibitor on arg1+ve exosomes rather than suggesting exclusivity for TAM-derived ones.
